# Applications of 3D Reconstruction in Virtual Reality-Based Teleoperation: A Review in the Mining Industry

**Alireza Kamran-Pishhesari, Amin Moniri-Morad and Javad Sattarvand \***

Department of Mining and Metallurgical Engineering, University of Nevada, Reno, 1664 N. Virginia St., Reno, NV 89557, USA; akamranpishhesari@unr.edu (A.K.-P.); amonirimorad@unr.edu (A.M.-M.)
\* Correspondence: jsattarvand@unr.edu

**Abstract:** Although multiview platforms have enhanced work efficiency in mining teleoperation systems, they also induce "cognitive tunneling" and depth-detection issues for operators. These issues inadvertently focus their attention on a restricted central view. Fully immersive virtual reality (VR) has recently attracted the attention of specialists in the mining industry to address these issues. Nevertheless, developing VR teleoperation systems remains a formidable challenge, particularly in achieving a realistic 3D model of the environment. This study investigates the existing gap in fully immersive teleoperation systems within the mining industry, aiming to identify the most optimal methods for their development and ensure operator's safety. To achieve this purpose, a literature search is employed to identify and extract information from the most relevant sources. The most advanced teleoperation systems are examined by focusing on their visualization types. Then, various 3D reconstruction techniques applicable to mining VR teleoperation are investigated, and their data acquisition methods, sensor technologies, and algorithms are analyzed. Ultimately, the study discusses challenges associated with 3D reconstruction techniques for mining teleoperation. The findings demonstrated that the real-time 3D reconstruction of underground mining environments primarily involves depth-based techniques. In contrast, point cloud generation techniques can mostly be employed for 3D reconstruction in open-pit mining operations.

**Keywords:** 3D reconstruction; teleoperation; mining industry; virtual reality

## 1. Introduction

Teleoperation in the mining industry has gained massive attention due to the safety and health issues associated with this operation. This has led operators to perform tasks from a safer distance, reducing exposure to hazardous environments. Furthermore, the increasing operational costs in the mining industry have led companies to adopt innovative technologies. This integration not only improves safety but also streamlines operations, enhances overall production efficiency, and boosts reliability [1,2]. Implementing driverless vehicles in mining operations for haulage improves safety by eliminating the need for human operators to navigate hazardous working environments, minimizing the risk of accidents, and ensuring a secure and controlled mining operation [3,4]. However, ever-changing factors affecting mining operations and the complexity of rock loading/excavation hinder the widespread utilization of fully autonomous systems in underground and surface mines. By integrating human oversight with autonomous systems, a more reliable operational framework is established, capable of effectively addressing the dynamic and intricate demands of mining operations [5–7]. An efficient teleoperation system can significantly reduce the number of hazards in mine sites by removing the operators and workers from dangerous working environments.

The historical analysis of the National Institute for Occupational Safety and Health (NIOSH) and the Mine Safety and Health Administration (MSHA) databases [8,9], spanning from 2000 to 2022, as depicted in Figure 1, reveals that despite the implementation of diverse

safety measures in United States mining operations, inherent risks persist. The analysis indicates a decline in the number of injuries from 1630 in 2000 to 5049 in 2022; however, it is apparent that the figures remain notably significant.

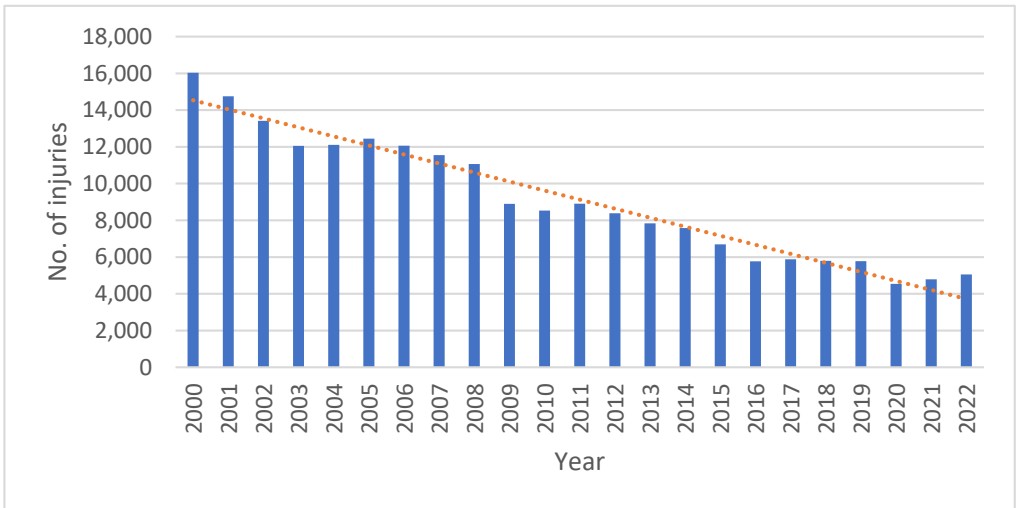

**Figure 1.** The number of injuries, including fatalities in mining operations, from 2000 to 2022.

A substantial factor contributing to mining accidents is powered haulage, presenting considerable risks to both operators and laborers. The movement of haulage units, including trucks and conveyors, introduces inherent dangers [10]. Despite advancements in safety measures, regulations, and the integration of proximity detection technologies, our data analysis from 2000 to 2022, as depicted in Figure 2, reveals a significant number of five fatalities in 2022.

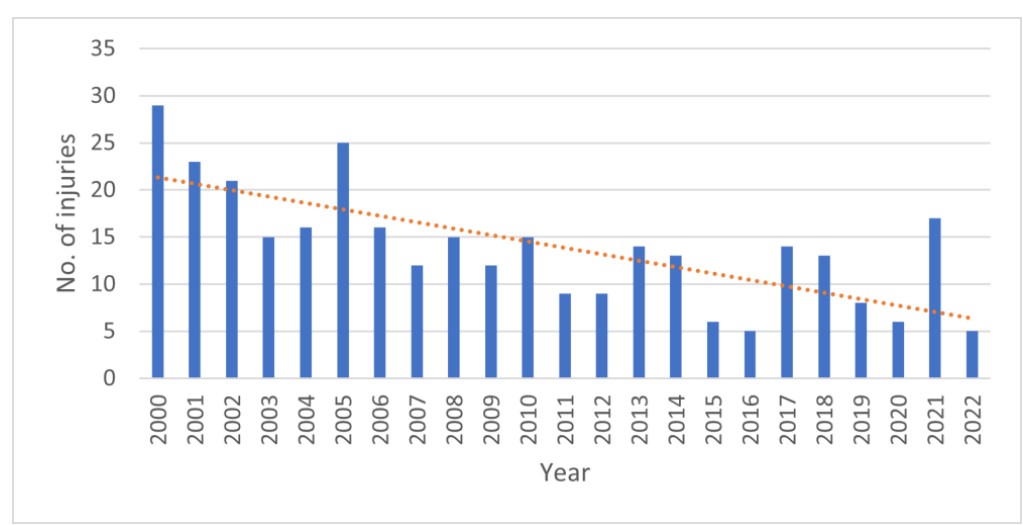

**Figure 2.** Number of powered haulage fatalities in mining operations from 2000 to 2022.

While powered haulage constitutes a relatively small share of all injuries (with an annual average of 8% of total injuries), it consistently plays a significant role in incidents leading to death or severe harm, in conjunction with other contributing factors. A comparative classification of fatality data from 2000 to 2022, as illustrated in Figure 3, indicates that powered haulage accounts for 32% (320 out of 983) of the fatalities during this period. This procedure analyzed factors that resulted in more than 50 fatalities during the specified timeframe.

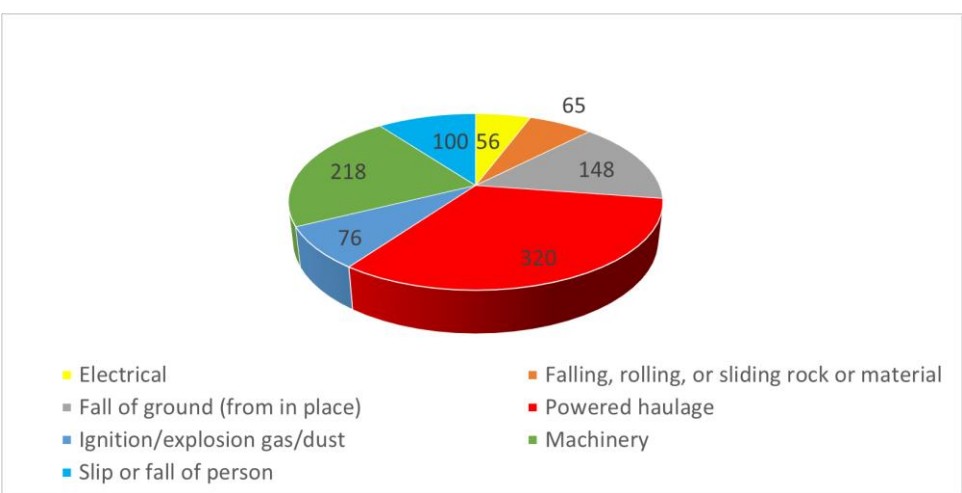

**Figure 3.** The number and classes of fatalities occurred from 2000 to 2022.

Teleoperation systems can be one of the most promising solutions to address these challenges, but they are not widely used because of low efficiency. One of the most critical factors influencing the efficiency of teleoperation is the operator's awareness of the working environment, also known as situational awareness [5,11]. Enhancing operators' situational awareness can be achieved through various tools: furnishing a comprehensive map of the working environment, ensuring a real-time and synchronized stream of videos or amalgamated data, providing precise spatial information, and strategically reducing the number of monitors to mitigate cognitive tunneling [12,13]. Creating a 3D model of the operational environment constitutes a significant method to augment visualization and situational awareness [14]. Nevertheless, these approaches are constrained by their high computational costs [12].

In recent years, significant strides in 3D construction techniques have facilitated the precise, non-real-time reconstruction of static scenes. This is achieved through methods such as depth map fusion, which involves employing multiple RGB-D cameras [15] alongside point cloud-generating sensors like Light Detection and Ranging (lidar) [16,17]. However, the challenge persists in developing a real-time solution capable of concurrently reconstructing two diverse mining environments with the requisite accuracy for teleoperation applications, necessitating further research [18]. Mining environments are mainly categorized into two major groups, surface and underground; similar to outdoor and indoor settings for 3D reconstruction, they present a mix of static and dynamic scenes encompassing rigid and non-rigid objects. This diverse array of environmental factors poses a multifaceted challenge in achieving real-time 3D reconstruction meeting the demands of teleoperation systems.

Contemporary teleoperation systems encompass three primary components: (1) the robot integrated with various sensors, (2) the communication infrastructure encompassing servers and algorithms, and (3) the operator's user interface, which can take the form of a monitor or Head-Mounted Display (HMD) [19]. The effectiveness of 3D reconstruction techniques utilized in these systems is contingent upon the specific environmental factors, such as indoor or outdoor settings, the nature of 3D reconstruction (real-time or non-real-time), the characteristics of the objects (rigid or non-rigid), the object's motion (static or dynamic), and the relative positioning of the camera and the object. Diverse methodologies cater to these conditions and requirements in the 3D reconstruction process.

The primary objective of this review study is to bridge the disparity between present-day teleoperation systems and the potential application of VR in mining operations. Given the inherent hazards of mining activities, ranging from rock falls to dusty environments and equipment collisions, it becomes imperative to explore the viability of integrating 3D reconstruction techniques in current teleoperation systems to enhance the operator's

situational awareness. To address these challenges, this study conducts a systematic review, examining various facets of novel platforms, sensors, and algorithms pertinent to 3D reconstruction. The intended contributions of this study encompass the following:

- The identification and evaluation of existing teleoperation platforms capable of potential transformation into VR-based teleoperation systems tailored for mining operations.
- The investigation and classification of sensor technologies that possess the capacity to enhance telepresence within the mining operation.
- The analysis of 3D reconstruction research studies and algorithms relevant to scenarios adaptable for employment in surface and underground mining telepresence scenarios.

The paper is structured as follows: Section 2 outlines the systematic review methodology, emphasizing criteria for selecting teleoperation platforms and research papers. Section 3 offers a detailed explanation of existing teleoperation platforms and. discusses 3D reconstruction methods for enhancing teleoperation systems in the mining industry. Section 4 addresses the limitations of current teleoperation systems and proposes leveraging 3D reconstruction for enhanced operator awareness in virtual reality. Finally, Section 5 concludes by emphasizing the underutilized potential of 3D reconstruction in mining teleoperation and anticipating long-term benefits in safety and productivity.

## 2. Materials and Methods

This systematic review study comprises two key phases, as depicted in Figure 4, to address aspects of teleoperation systems and relevant research studies comprehensively.

As shown in Figure 4, Phase 1 focuses on evaluating the existing landscape of teleoperation platforms, identifying those adaptable to fully immersive VR systems for mining operations. This phase adopts a comprehensive approach, drawing insights from both research laboratories and industry, involving an in-depth analysis of selected teleoperation platforms. Nineteen teleoperation systems from research laboratories ($n = 19$) and nine systems from the industry ($n = 9$) were chosen for the study. Phase 2 involves a meticulous analysis, emphasizing relevant research studies on 3D reconstruction techniques applicable to surface and underground mining teleoperation scenarios. This phase includes the analysis of sensor technologies' capacity to enhance teleoperation in the mining sector (Phase 2(a)), along with extracting and elucidating algorithms and sensors employed in selected research studies (Phase 2(b)), highlighting their applications and relevance in the context of mining teleoperation.

Research has been conducted by investigating and screening companies' websites, handbooks, technical reports, and papers from IEEE Xplore, Science Direct, and Springer Link using keywords such as "Real-time, Non-real-time 3D reconstruction, visualization", "Indoor, outdoor 3D reconstruction, visualization", and "Dynamic scene, static scene". Since this survey contains both mining environments, research papers from other industries are also included because of their similarity. The screening was conducted by considering multiple criteria, including date, language, technological aspects, and fields of study. At the first step, 458 papers ($n = 458$) were selected for the review. The next step involved title and keyword screening to eliminate papers not focused on 3D reconstruction. It was followed by abstract screening, removing papers unsuitable for real-time indoor 3D reconstruction and non-real-time outdoor reconstruction or lacking teleoperation potential. Full-text screening identified and excluded papers unfit for application in mining VR immersive teleoperation. The quantity of selected publications concerning the respective publication years, depicted in Figure 5, illustrates that most relevant studies in the 3D reconstruction of indoor environments (32 studies) were conducted between 2020 and 2022. The increase in research is linked to the prevalent accessibility of reasonably priced RGBD sensors, exemplified by Microsoft's Kinect, delivering precise depth details in conjunction with color data. Progress in computational capabilities, especially in GPUs, eased the real-time processing of extensive datasets. Incorporating machine learning and computer vision methodologies and a rising enthusiasm for AR and VR applications amplified the need for precise and effective indoor mapping.

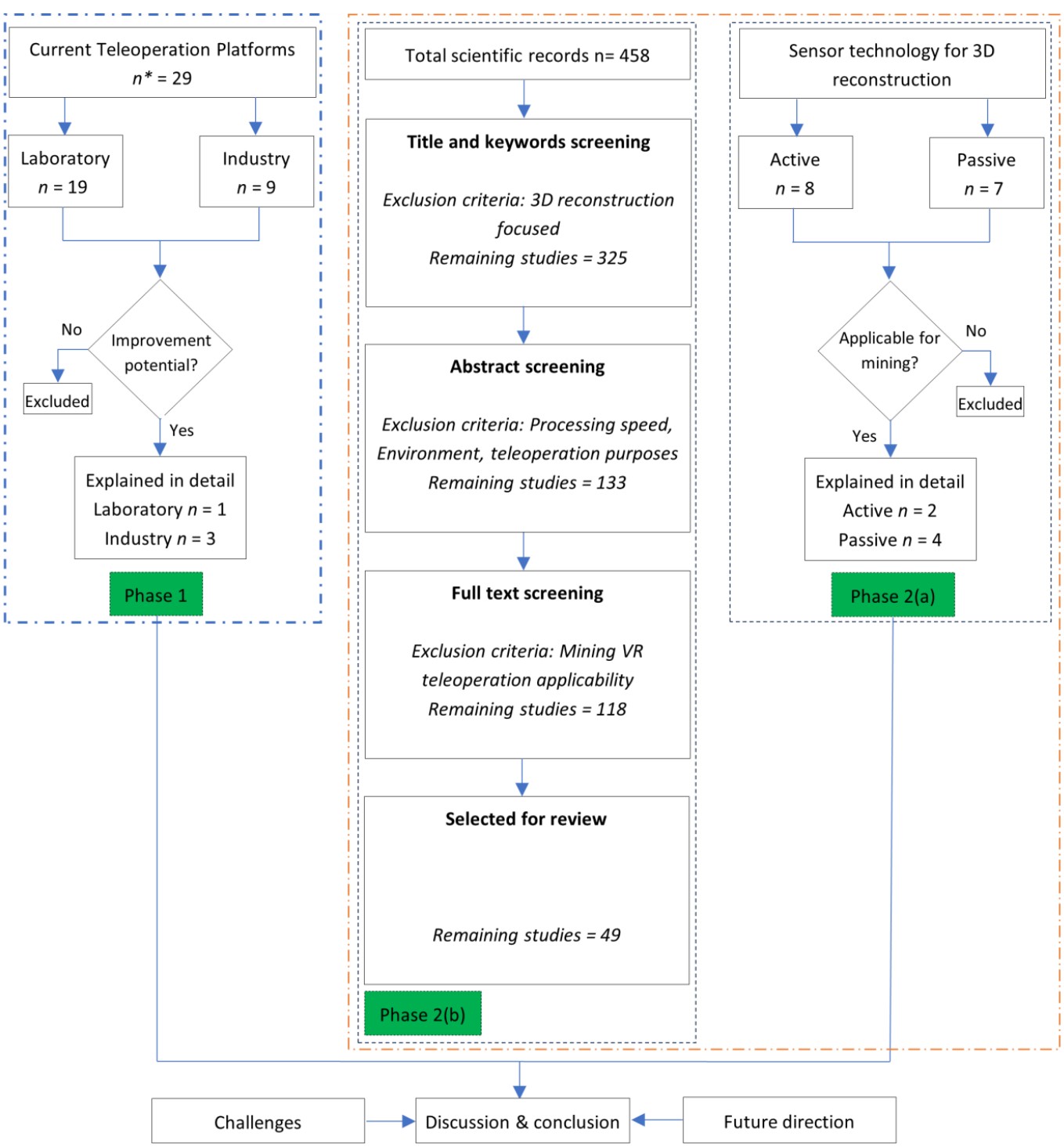

**Figure 4.** The main body of the research.

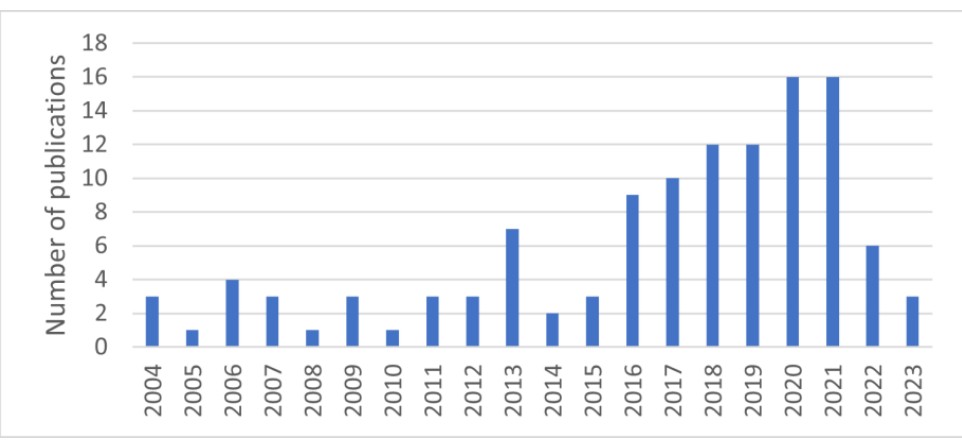

**Figure 5.** The number of publications per year.

## 3. Results

The Materials and Methods Section involved a comprehensive evaluation of teleoperation platforms for mining VR systems, resulting in selecting one laboratory and three industry teleoperation systems. A rigorous screening of 458 papers through title, keyword, abstract, and full text identified 49 pertinent papers. Additionally, this process identified six sensors, further enriching the study with a focused exploration of sensor technologies suitable for mining VR immersive teleoperation.

### 3.1. Current Technologies in Teleoperation

In recent years, it has been possible to teleoperate mining equipment efficiently thanks to massive technological achievements in sensor production, 5G communication networks, visualization tools, and algorithms. Two crucial sections should be considered when investigating the state-of-the-art mining and construction teleoperation systems technologies. The first sector comprises programmable teleoperation systems, predominantly developed within laboratories for research purposes. The second segment encompasses equipment manufacturer companies dedicated to constructing and commercializing these teleoperation platforms.

#### 3.1.1. Research Laboratories

One of the most advanced teleoperation systems has been developed as a remotely operated walking excavator [20] research project called HEAP [21] (a customized Menzi Muck M545 Excavator [22]), indicated in Figure 6, in robotic labs at ETH Zurich. This platform was first presented at Bauma Fair 2019 [23].

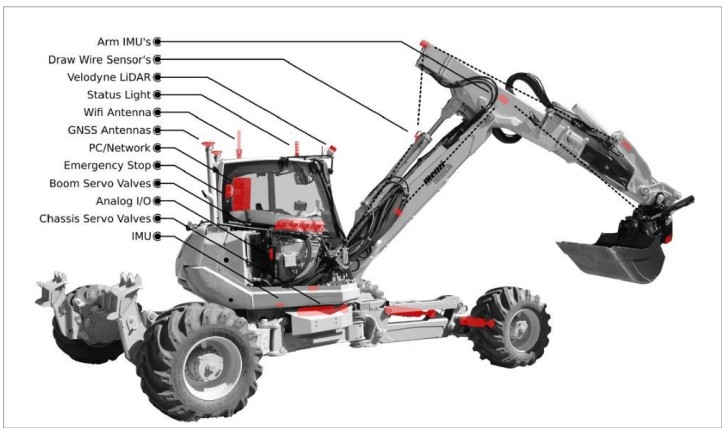

**Figure 6.** Sensorized custom Menzi Muck M545 robot called HEAP [24].

A multicamera vision system installed on the excavator uses industrial cameras from XIMEA, which utilize low-latency image processing provided by MRTech SK (MRTech™) City: Western Slovak city Country: Slovakia [25] and run on the NVIDIA Jetson TX2 platform. Within the HEAP platform, the cockpit is simulated atop a 3-degree-of-freedom (3-DoF) motion platform, indicated in Figure 7. This setup ensures synchronized motion between the chassis and the platform during excavator operation. The initial iteration of IBEX [26] provided visual feedback through three 3D monitors displaying live feeds from three distinct cameras. The primary use of this platform is for construction purposes, but it also has the potential to be employed in mining operations.

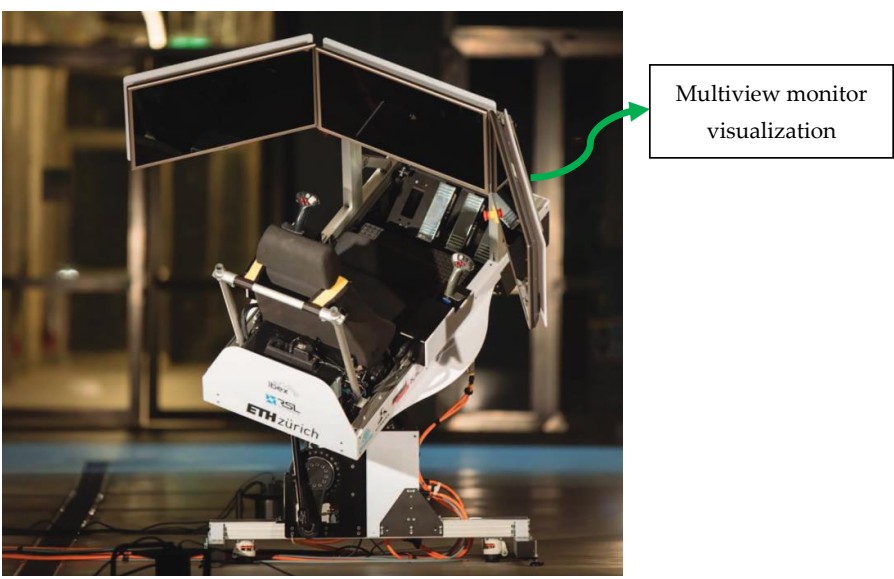

**Figure 7.** IBEX motion platform [26].

### 3.1.2. Heavy Equipment Manufacturers

The pinnacle of advanced teleoperation technology is evident in heavy equipment manufactured by industry-leading companies such as Komatsu, Caterpillar, Hitachi, Sandvik, Epiroc, Liebherr, Doosan, Volvo, and Hyundai. Despite these advancements, these platforms continue to rely on multiple monitors to visualize the operational environment.

As the most advanced technology of Komatsu in Teleoperation, a new PC-7000 excavator was teleoperated, loading a fully autonomous concept truck in Arizona from Las Vegas during MINExpo 2021 [27]. An Immersive Technologies [28] console provided the information and visualization of the working environment separated by 413 miles. The components of the teleoperation system used in the platform include the following:

- A 360° view monitor and machine display;
- The implementation of a semi-automated loading system;
- Real-time operator guidance and coaching through augmented reality (AR) technology.

Doosan is the pioneer in deploying 5G technology for the global "Teleoperation" of construction machinery, debuting this groundbreaking platform at the Bauma 2019 exhibition in Munich, Germany. The innovation was demonstrated by operating a Doosan DX380LC-5 40-tonne crawler excavator over 8500 km away in Incheon, South Korea, from the Bauma stand in Munich. The teleoperation system used a low-latency video transmission module and rapid video transfer through encoding and decoding methods, which is crucial in minimizing time delays [29,30].

Caterpillar has introduced an advanced technology known as "Cat® Command," a versatile platform offering remote control, semi-autonomous, and fully autonomous functionalities for equipment operations [31]. This platform includes two variants tailored for teleoperation: a console and a station. The console is designed to enable operators to control the machinery near the site [32], which is available for select Cat dozers, excavators,

and wheel loaders. The Cat Command Station, available for Cat dozers, excavators, and wheel loaders, is designed to enable operators to conduct operations remotely [32,33]. The Cat Command system has been adapted for underground operations, offering three distinctive levels of operation [34]:

- "Teleremote"—employed for bucket loading and unloading;
- "Copilot"—a semi-autonomous mode;
- "Autopilot"—facilitating autonomous machine operation.

Mentioned technologies use multiple monitors to visualize the working environment. However, there is a severe problem with this type of visualization called "cognitive tunneling". This means that providing operators with excess visual information makes them subconsciously pay attention to a limited view, thereby ignoring other views. In addition, depth detection is not possible using 2D monitors.

*3.2. 3D Reconstruction Approaches for Immersive Mining Teleoperation*

Integrating 3D reconstruction techniques holds considerable potential for enhancing immersive teleoperation in mining operations [35]. The initial stage in 3D reconstruction for immersive mining teleoperation involves exploring diverse data acquisition options, including active and passive sensors, and selecting the most suitable one based on the type of challenges and requirements of the mining environment. This careful selection lays the foundation for capturing accurate spatial information, which is essential for creating detailed 3D models. 3D modeling in surface mines differs significantly from underground mines due to distinct environmental challenges. In surface mining, sensors like lidar and RGB cameras are often employed to capture large-scale structures and detailed surface features effectively. However, in underground mines with limited visibility, low light, and dust, sensors such as RGB-D cameras and lidar become crucial, offering real-time capabilities and depth sensing. The 3D reconstruction of surface and underground mines diverges in sensor selection and applied algorithms. In surface mining, structure from motion (SfM) techniques, leveraging cameras and photogrammetry, are commonly utilized, as indicated in Figure 8, created by Metashape software [36]. These methods excel in reconstructing large-scale surfaces, incorporating multiview stereo (MVS) vision and simultaneous localization and mapping (SLAM) for detailed reconstructions. Conversely, underground mining demands algorithms capable of navigating low-light conditions, limited visibility, and the presence of dust. Simultaneously, algorithms like SLAM and volumetric methods, accommodating the unique challenges of underground environments, take precedence.

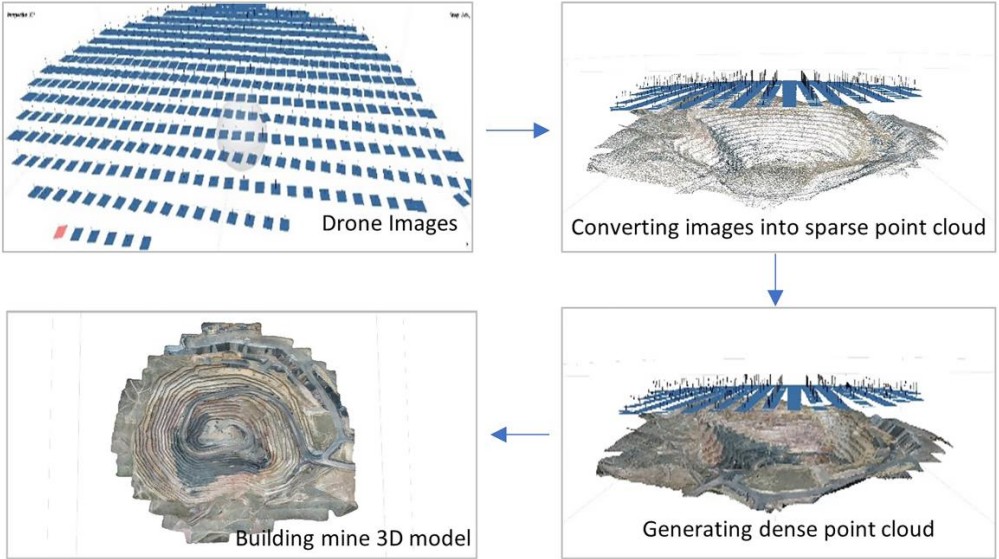

**Figure 8.** The steps of creating a 3D model of a surface mine using images.

### 3.2.1. Data Acquisition for 3D Reconstruction

In recent years, various sensor technologies have been pivotal in 3D reconstruction, capable of directly or indirectly measuring point clouds. A point cloud refers to a set of data points in a 3D coordinate system, often characterized by X, Y, and Z coordinates, typically representing the external surfaces of objects. In addition to mapping 3D coordinates, point clouds can encompass attributes such as intensity, RGB color, impulse return numbers, and semantic information, providing more in-depth details. The measurement of point clouds is typically accomplished through two primary data acquisition methods [37], i.e., range [38] and image [39,40], known as range-based and image-based approaches, respectively. Sensors employed for point cloud measurement can be categorized into two primary groups: active and passive. Active sensors directly collect the point cloud data, while passive sensors, such as different cameras, capture images that contribute to the formation of point clouds [37,41].

#### Active Sensors

Active sensors proactively emit signals or energy into the environment to gather information. These sensors independently generate and transmit signals, allowing them to interact actively with the surroundings for data collection.

- Lidar

The term lidar, presented in Figure 9a [42], derived from Light Detection and Ranging, characterizes a technology essential for active remote sensing, relying on the time between emitting and receiving laser signals reflected from a target to measure distances [43,44]. Pulsed lasers are predominantly utilized in civil and mining activities, primarily in terrestrial applications.

The position and type of lidar sensors play a crucial role in determining the density of acquired point clouds. Airborne lidar scanning achieves decametric resolutions, while ground-based laser scanning, known as terrestrial laser scanning (TLS), achieves centimetric to millimetric resolutions. Another terrestrial approach, known as Terrestrial Mobile Laser Scanning (TMLS), enables point cloud collection from moving vehicles and is adaptable to both indoor and outdoor environments [43,45].

Terrestrial and, more recently, airborne laser scanning, notably through drones equipped with lidar sensors, are extensively utilized in close-range methods for 3D reconstruction in both surface and underground mining operations. These methods prove essential for tasks such as discontinuity mapping for slope monitoring, a critical safety measure [46].

- Radar

The term radar, derived from radio detection and ranging, represents a detection system that employs radio waves to detect objects and compute targets' range (distance), angle, and velocity [47]. A combination of radar with a vision sensor, indicated in Figure 9b, has the potential for 3D reconstruction applications [48]. The robustness of radar in outdoor environments makes it suitable as a reliable 3D reconstruction sensor for surface mines.

#### Passive Sensors

These sensors rely on naturally occurring stimuli, such as ambient light or thermal radiation, to gather information from their surroundings. Passive sensors observe and respond to environmental conditions without initiating external energy transmission [49].

- RGB-D Camera

These sensors are specialized devices designed for depth sensing in conjunction with an RGB camera, generating image outputs embedded with depth information, showcasing the significant potential for indoor 3D reconstruction, particularly when paired with simultaneous localization and mapping (SLAM)-based algorithms [50]. These sensors, indicated in Figure 9c, produced by various manufacturers such as Microsoft, Asus, and Intel, among others, are notably affordable and have established themselves as reliable tools

for research purposes in 3D modeling [51]. However, they present critical challenges related to limited range and degradation of depth accuracy, prompting extensive research into sensor calibration to enhance precision. Typically, these sensors comprise an RGB camera, an IR (infrared) camera, an IR projector, and an IR source [52]. Despite their advantages, utilizing RGB-D cameras in an underground mine environment presents notable challenges because of dust and low illumination.

- Monocular Camera

Monocular cameras, indicated in Figure 9d [53], a prevalent type of vision sensor, differ from binocular cameras as they consist of a single lens. Traditionally employed for object detection due to their 2D image (or video) output, these sensors were unsuitable for depth estimation [54]. They are specifically engineered for applications requiring compact, lightweight, and potentially cost-effective cameras [55]. Many deep learning methods have emerged for depth estimation and 3D reconstruction in underground and surface environments [56].

- Binocular/Stereo Camera

A stereo camera, presented in Figure 9e [57], comprising two monocular cameras, utilizes the parallax principle to estimate scene depth, providing a tool to obtain a dense depth map from images [55]. A dense depth map can be obtained from images [58]. Leveraging structure from motion techniques, stereo cameras find extensive application in 3D reconstruction in surface environments. Nonetheless, the calibration process for these paired cameras presents notable challenges [59].

- Fisheye Cameras

Fisheye cameras, a variation of monocular cameras presented in Figure 9f [60], boast an ultra-wide angle of view, making them particularly adept for object detection [55]. Their expansive field of view (FOV) renders them suitable for large-scale 3D reconstructions in surface mines, capturing a more comprehensive perspective than other cameras. Alternatively, employing deep learning techniques for depth extraction, akin to monocular 3D reconstruction, is a viable approach [61].

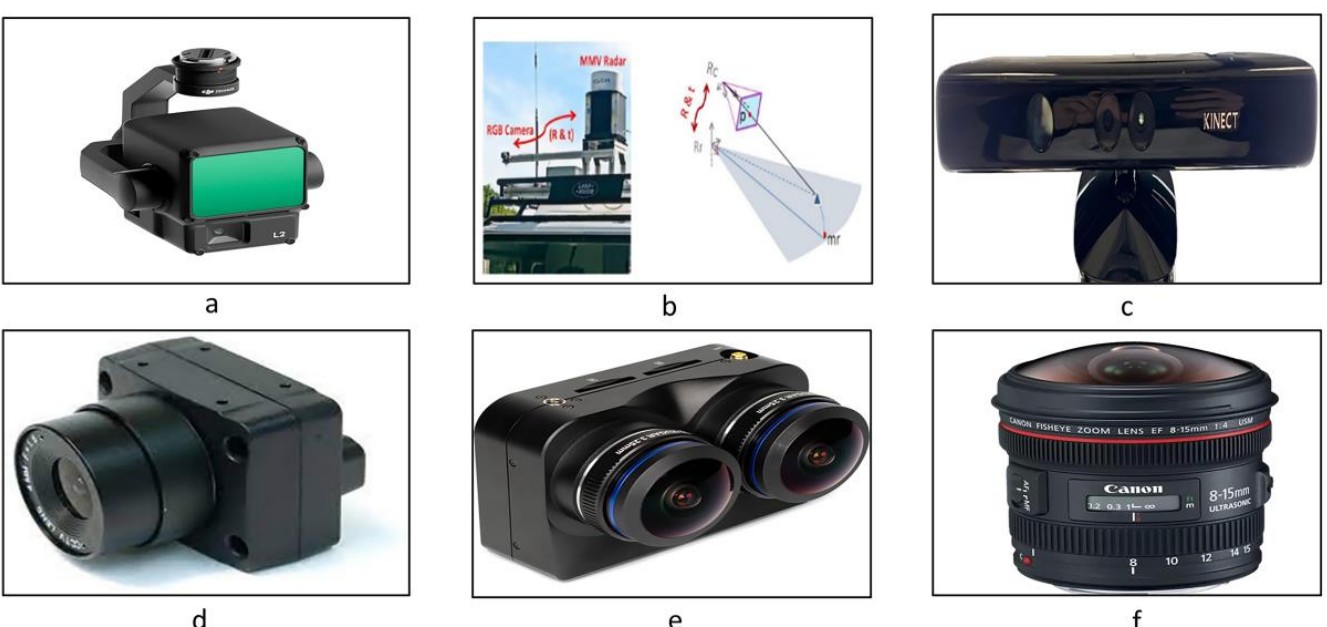

**Figure 9.** Lidar sensor (**a**) [42], radar (**b**) [48], RGB-D camera (**c**), monocular camera (**d**) [53], binocular (**e**) [57], and fisheye (**f**) [60].

Table 1 comprehensively evaluates various sensors—lidar, radar, RGB-D camera, monocular camera, stereo camera, and fisheye camera—focusing on their applicability to 3D reconstruction in mining environments. The columns describe the essential criteria: range, accuracy, environment suitability, real-time capability, and robustness in dusty/noisy environments. This analysis aims to assist in selecting the most suitable sensors for mining applications based on their performance in these key aspects. According to the table, lidar emerges as the top-performing sensor, boasting a long-range (up to 300 m+) and high accuracy at the millimeter level for both environments. The moderate real-time capability can be improved using high-performance processing hardware. It exhibits robustness in dusty/noisy conditions, making it a comprehensive choice. Following closely is the RGB-D camera, which is suitable for underground environments and has high real-time capability. While not explicitly specified for surface mine environments, the RGB-D camera offers versatility in capturing detailed 3D data. On the contrary, monocular cameras lag as the less favorable option due to their limited accuracy and range. It is essential to note that the comparison of sensors is based on general characteristics, and the performance can vary significantly depending on the specific models.

**Table 1.** A comparison of sensors based on their application for 3D reconstruction in mining.

| Sensor | Range | Accuracy | Environment Suitability | Real-Time Capability | Robustness in Dusty/Noisy Environments |
|---|---|---|---|---|---|
| Lidar | Long (up to 300 m+) | High (mm level) | Underground and surface | Moderate to high | Moderate to high |
| Radar | Medium to long | Moderate to high | Underground and surface | High | High |
| RGB-D | Short to medium | Moderate to high | Underground | High | Low to moderate |
| Monocular | Short | Low to moderate | Underground and surface | High | Low |
| Stereo | Short to medium | Moderate to high | Underground and surface | Moderate to high | Moderate to high |
| Fisheye | Wide field of view | Low to moderate | Underground and surface | High | Low to moderate |

### 3.2.2. 3D Reconstruction in Surface Mines

In surface mining applications, the generation of 3D reconstructions primarily relies on point clouds derived from a mix of various RGB cameras (monocular, binocular, stereo), laser scanners, or a hybrid approach [62,63]. Several methodologies contribute to creating point clouds with 3D measurements, including multiview stereo (MVS) vision, structure from motion (SFM), simultaneous localization and mapping (SLAM), and single image depth estimation.

Of these approaches, the "structure from motion" method, illustrated in Figure 10, is being investigated in this section due to its extensive functionality in reconstructing mining environments. This method encompasses various stages, including camera calibration, feature extraction, feature matching, sparse 3D reconstruction, model parameter correction, and dense 3D reconstruction [64,65]. The steps involved in this process can vary depending on the camera type.

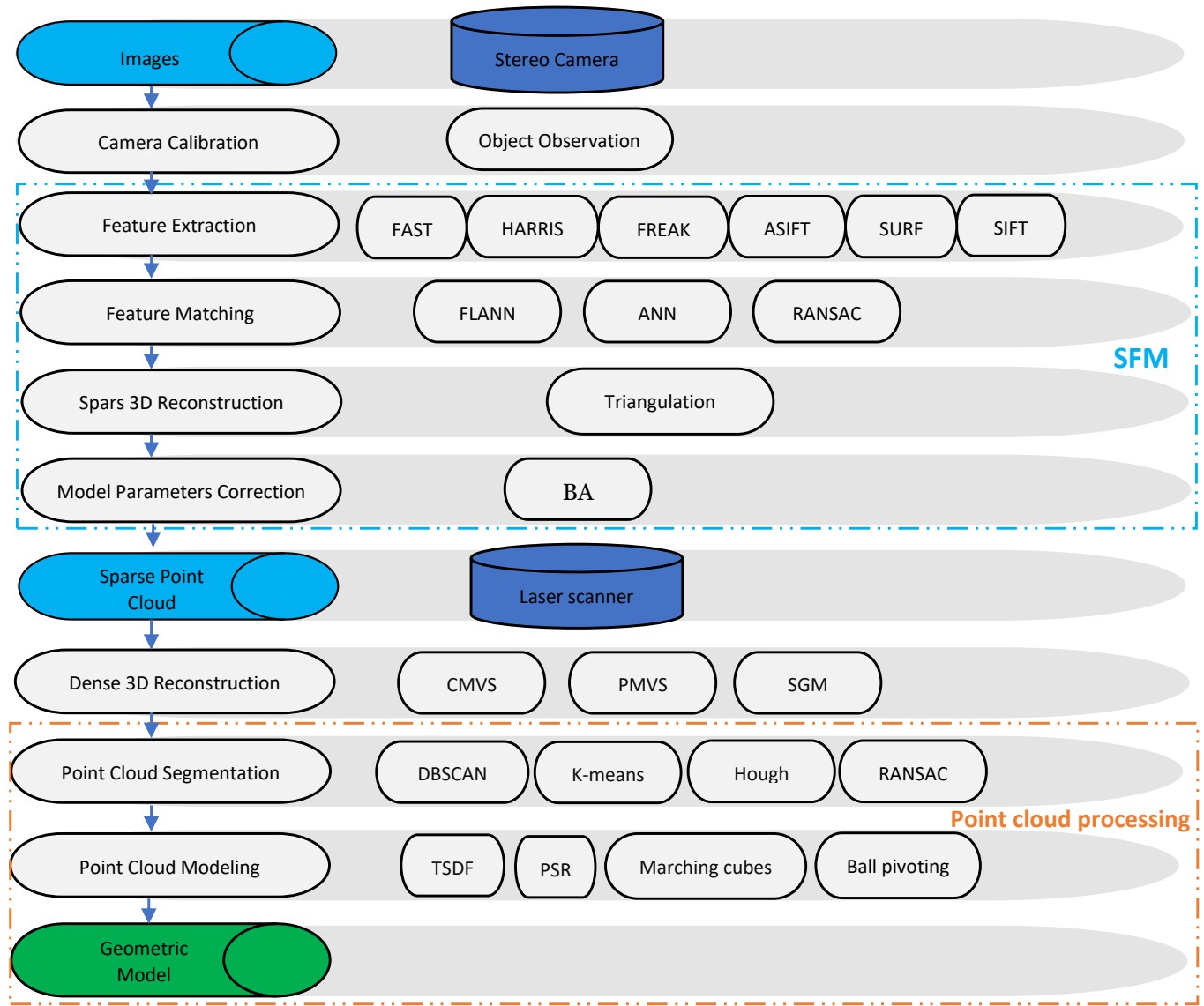

**Figure 10.** The procedure of constructing geometry models from images.

Point Cloud Generation from Stereo Images

In generating point clouds from stereo images, the process typically involves a sequential five-step approach: camera calibration, feature extraction, feature matching, sparse 3D reconstruction, model parameter correction, and dense 3D reconstruction. Each stage's output serves as the input for the subsequent step, signifying that the accuracy of the resulting 3D model significantly hinges on the initial processing phases. The effectiveness and precision at these early stages are pivotal in achieving a high-quality final 3D model.

Stereo Camera Calibration

Camera calibration involves estimating intrinsic and extrinsic camera parameters, lens distortion coefficients, etc. Two primary calibration methods are utilized. Chart-based calibration (CBC) relies on an object with known geometry in all input images and structures from motion. Intrinsic camera parameters are crucial before the process or can be recovered a posteriori through auto-calibration [66]. A bundle adjustment (BA) stage refines the positions of points and all parameters [67]. Selecting feature correspondences (SFC) is pivotal in both methods. Stereo camera calibration works to calibrate and calculate intrinsic and extrinsic parameters for both lenses [68–70]. First, intrinsic parameters for each

camera are estimated using pre-measured spatial geometry. Then, extrinsic parameters are estimated via SfM and refined by patch-based multiview stereo and bundle adjustment [65].

Feature Extraction

The objective of this step is to collect feature points from images, which is achieved through two primary types of algorithms: feature point detectors and descriptors. Detector algorithms determine the point locations, which are then characterized by descriptor algorithms. Various algorithms, including SIFT (scale-invariant feature transform), ASIFT (affine scale invariant feature transform), SURF (speeded-up robust features), FAST (features from accelerated segment), ANMS (adaptive non-maximal suppression), and Forstner–Harris, are commonly used for detection due to their robustness [71]. For instance, in SIFT, a scale space is created by convolving a primary image with a Gaussian function, generating a series of images. The difference-of-Gaussian images are obtained by eliminating neighboring images from this space. Point candidates are gathered by differentiating adjacent pixels, and final feature points are selected via candidate filtering [72]. In ASIFT, all possible distortions stemming from the camera's optical axis orientation are considered, producing reproduced images of the primary image. Subsequently, SIFT is employed to collect feature points [73].

SURF detects feature points by approximating the Hessian determinant of pixels in the scale space and selecting candidates based on these approximations [74]. Harris corner detection assesses intensity changes around each pixel to identify corners [75]. FAST identifies corner pixels based on pixel count within a specific circle of neighboring pixels, determining corners by exceeding a set threshold [76]. While most detectors search for local maximums, ANMS selects only the maximum features that significantly surpass adjacent features within a set radius, preventing uneven distribution [71,77].

For feature description, various descriptors are employed, i.e., SIFT, Surf, FREAK (Fast REtinA Keypoint), and MSD (Multi-Scale Descriptor). The SIFT descriptor calculates gradient magnitude and orientation for each image point within a defined radius to classify feature points [72]. Surf generates vectors using Haar-wavelet responses [78]. FREAK creates a binary string pattern based on one-bit differences of Gaussians [79]. MSD utilizes intensity gradients over multiple scales for vector generation to classify feature points [72].

Table 2 assesses prominent feature detection algorithms—FAST, Harris, ASIFT, SURF, and SIFT—specifically tailored to their applicability in 3D reconstruction within mining environments. Each algorithm is scrutinized based on essential criteria outlined in the columns: application, process speed, and robustness to scale of the images. The table reveals that selecting the "best" and "worst" feature extractor from the table requires a nuanced consideration of specific application requirements. FAST, specializing in keypoint detection, exhibits high process speed and robustness to scale, making it a strong candidate for real-time 3D reconstruction of surface mines. SURF offers a balance between speed and robustness.

**Table 2.** A comparison of feature extractors in SFM.

| Feature Detectors | Application | | | | Process Speed | | Robustness to Scale | |
|---|---|---|---|---|---|---|---|---|
| | Object Recognition | Image Matching | Image Stitching | Keypoint Detection | High | Mod * | Robust | Limited |
| FAST | - | - | ✓ | ✓ | ✓ | - | - | ✓ |
| Harris | ✓ | ✓ | - | - | ✓ | - | - | ✓ |
| ASIFT | ✓ | - | ✓ | - | - | ✓ | ✓ | - |
| SURF | ✓ | ✓ | - | - | ✓ | - | ✓ | - |
| SIFT | ✓ | - | ✓ | - | - | ✓ | ✓ | - |

* Mod stands for moderate.

Feature Matching

Feature matching from multiple images is critical to identify and discard incorrect matches. Approximate Nearest Neighbor (ANN) algorithm initially matches feature points using its Euclidean distances, possibly leading to incorrect matches. The Fast Library for Approximate Nearest Neighbor (FLANN) is used as an alternative matching algorithm. Random Sample Consensus (RANSAC) is applied to eliminate false matches. RANSAC estimates epipolar geometry by randomly sampling feature points, effectively reducing false matches. Outlier Removal by Sequential Analysis (ORSA) performs better than RANSAC when incorrect matches exceed a specific threshold.

Table 3, comparing the feature matchers in SFM regarding their processing speed and robustness to the scale of the images, uncovers that FLANN and ANN, both utilizing approximate nearest neighbor techniques, showcase high process speed and moderate robustness to scale are suitable for applications prioritizing quick processing such as real-time 3D modeling of the surface mines.

**Table 3.** A comparison of the feature matchers in SFM.

| Feature Matcher | Process Speed | | Robustness to Scale | |
| --- | --- | --- | --- | --- |
| | High | Med | Mod | Limited |
| FLANN | ✓ | - | - | ✓ |
| ANN | ✓ | - | - | ✓ |
| RANSAC | - | ✓ | ✓ | - |

Sparse 3D Reconstruction

This step aims to create a sparse point cloud using triangulation algorithms. Data from the camera parameters calculated during camera motion estimation and matched image pairs from the previous step are essential to generate the point cloud. The point cloud is generated by computing the 3D coordinates of the points.

Model Parameter Correction

To refine camera parameter correction, the 3D locations of points from the previous step and estimated camera parameters are utilized. Bundle adjustment, employing a non-linear least square technique, is a robust approach for optimizing the 3D point locations and camera parameters.

Dense 3D Reconstruction

In subsequent stages, the obtained camera parameters and sparse point clouds are refined to create a denser, more precise point cloud. However, errors can accumulate during this process, necessitating the elimination of unnecessary images. This is achieved through CMVS (clustering views for multiview stereo) to cluster images. Subsequently, PMVS (patch-based multiview stereo) utilizes the final images to produce a denser point cloud. Another method for achieving denser point clouds is SGM (Semiglobal Matching), which optimizes a global energy process.

A comparison of dense 3D reconstruction algorithms in Table 4 regarding their scalability, accuracy, computational efficiency, applicability, and robustness based on different scenes indicates that PMVS is positioned as a strong contender, offering high accuracy in challenging scenes like surface mines with complex geometry and occlusions. Following closely is CMVS, known for its high scalability, accuracy in complex scenes, and efficiency, which are particularly suitable for reconstructing large-scale structures. While demonstrating versatility and generally robust performance, SGM holds a moderate position in terms of scalability and computational efficiency.

**Table 4.** A comparison of dense 3D reconstruction algorithms from sparse point clouds.

| Algorithm | Scalability | Accuracy | Computational Efficiency | Applicability | Robustness |
|---|---|---|---|---|---|
| CMVS * | High | High in complex scenes | Efficient | Large-scale structures | Complex scenes |
| PMVS ** | Depends on patch size | High in challenging scenes | Depends on patch size | Complex geometry and occlusions | Occlusions |
| SGM *** | Moderate | High in structured environments | Moderate (depends on scene size) | Versatile | Generally robust |

* CMVS stands for Clustering Views for Multiview Stereo, ** PMVS stands for Patch-based Multiview Stereo, *** SGM stands for Semiglobal Matching.

Point Cloud Segmentation

Segmentation of a dense point cloud into distinct regions/segments is crucial for various applications related to 3D reconstruction in mining environments. This section discusses several common techniques for processing point cloud data that are appropriate for 3D reconstruction:

1.  Voxel Grid Based: This technique divides the data into uniform cubes, known as voxels. It is versatile and widely applicable for analyzing, segmenting, and processing point cloud data, making it useful for object recognition, augmented reality, CAD modeling, and other applications in large-scale scenes [80].
2.  Plane Detection: Detecting flat surfaces (planes) in point cloud data is a versatile method in 3D reconstruction. RANSAC and Hough transformation are among this method's most common algorithms/techniques [81].
3.  Clustering: This method effectively clusters irregularly shaped objects, making it suitable for reconstructing outdoor environments. K-means and DBSCAN are popular algorithms used in this technique [82].
4.  Deep Learning Based: Convolutional neural networks have gained significant attention for their performance in 3D reconstruction. Techniques such as Mask R-CNN, FCN (Fully Convolutional Network), DeepLab, and PSPNet (Pyramid Scene Parsing Network) are notable in this domain [83].

A comparison of these algorithms in Table 5, regarding their accuracy, robustness, processing speed, and applications, reveals that RANSAC is a strong candidate due to its high accuracy in model fitting and robustness to outliers, which is needed to reconstruct outdoor environments such as surface mines. However, its processing speed is moderate to slow. DBSCAN, known for high accuracy in dense clusters and robustness to noise and outliers, is next, with its processing speed contingent on points and density.

**Table 5.** A comparison of point cloud segmentation algorithms.

| Algorithm | Accuracy | Robustness | Processing Speed | Common Applications |
|---|---|---|---|---|
| DBSCAN * | High in dense clusters | Robust to noise and outliers | Depends on points and density | Object recognition |
| K-means | Struggle with non-spherical or unevenly sized clusters | Sensitive to outliers | Fast, even for dense points | Clustering spherical structures |
| Hough Transform | High in detecting geometric shapes | Robust to noise | Computationally expensive | Line detection, circle detection |
| RANSAC ** | High in model fitting | Robust to outliers | Moderate to slow | Fitting models to geometric structures |

* DBSCAN stands for Density-based spatial clustering of applications with noise, ** RANSAC stands for Random Sample Consensus.

Point Cloud Modeling

This step involves modeling the segmented regions from the previous phase to generate accurate 3D visualizations of objects. Common methods used in this phase include the following:

1. Mesh Generation: Algorithms like Ball Pivoting and Marching Cubes are usually employed for this method [84].

2. Volumetric Based: Regular shape representation is commonly achieved using Voxels and TSDF (Truncated Signed Distance Function) [85].

The analysis in Table 6, regarding the accuracy, robustness, processing speed, and applications of the algorithms, uncovers that Marching Cubes emerges as the top choice due to its high accuracy across a diverse range of surfaces and adaptability to complex shapes and topologies, making it particularly well suited for terrain 3D modeling. Following closely is Ball Pivoting, recognized for its capacity to achieve high accuracy in reconstructing intricate structures and robustness in handling noisy data, albeit with a dependency on point cloud density.

**Table 6.** A comparison of point cloud modeling techniques.

| Algorithm | Accuracy | Robustness to | Processing Speed | Common Applications |
|---|---|---|---|---|
| TSDF * | High | Noisy input | Computationally intensive | 3D reconstruction in structured environments |
| Marching Cubes | High for a wide range of surfaces | Complex shapes and topologies | Depends on voxel resolution | Terrain 3D modeling |
| Ball Pivoting | High for intricate structures | Noisy data | Depends on point cloud density | 3D Reconstruction of irregular surfaces |

TSDF * stands for Truncated Signed Distance Function.

### 3.2.3. 3D Reconstruction in Underground Mines

Reconstructing highly detailed indoor environments poses considerable challenges due to limited visibility, complex object interactions, and occlusions that hinder complete scenes [86]. This complexity is exacerbated in underground mining conditions characterized by low light, leading to inaccuracies in reconstructing walls, roofs, floors, and 3D objects. Inadequate surface textures and the presence of dust further complicate accurate data acquisition. Addressing these challenges necessitates meticulous sensor selection and installation to navigate these environmental limitations [87]. Diverse sensors, including RGB, RGB-D, Lidar, various cameras, or their combinations, are suitable for indoor 3D reconstruction, each offering unique advantages based on the scene's requirements. The subsequent step involves algorithm selection tailored to the input data and the specific reconstruction type needed. This ensures the most effective generation of the 3D model based on the acquired data.

Real-time indoor 3D reconstruction algorithms can be classified based on their underlying methodologies. Some of the most used algorithms are simultaneous localization and mapping (SLAM) [88], volumetric methods [89], NN-based approaches [90], depth sensing and fusion [91], and hybrid methods [92].

These categories are not strictly separate, and specific algorithms may concurrently encompass features from multiple categories. The domain of 3D construction has seen extensive advancements over recent decades. The demand for 3D modeling is prevalent due to its vast applications in virtual and augmented reality, the gaming industry, robotics, fabrication, and numerous other fields. Additionally, the availability of high-quality and cost-effective sensors has significantly transformed this process [93].

An analysis of studies on real-time 3D reconstruction in indoor environments applicable for mining teleoperation is conducted in Table 7, consisting of a summary of 30 studies. Each row corresponds to individual research, while the columns encapsulate key characteristics and findings. The "Input Data" column categorizes the data type used in the studies,

distinguishing between RGB, RGB-D, and other sources. The "Methods" column classifies the general reconstruction techniques, including SLAM, volumetric, sparse, and hybrid approaches. The "Scene" column distinguishes between indoor and outdoor settings, providing insights into the environmental context of the studies. "Scalability" indicates whether the studies were conducted on a large, small, and even applicable for both scales. The subsequent columns delve into the robustness of the reconstruction methods, considering factors such as adaptability to reconstruct dynamic scenes or objects, resilience to deformable objects, and the specific algorithms or techniques employed in the last column.

According to the table, the RGB-D sensor emerges as the paramount choice for 3D reconstruction in underground mines, leading the field with 17 out of 30 studies in real-time reconstruction endeavors, followed by RGB sensors with 10 out of 30 studies. Lidar, IMU, and SWIR sensors were observed to be less frequently utilized in the surveyed studies. The availability of RGB-D sensors, coupled with the numerous sophisticated algorithms highlighted in the table, holds great promise in enhancing research efforts within this field, indicating a potential leap forward in 3D reconstruction methodologies.

Simultaneous localization and mapping (SLAM) stands out as the predominant method, featured in 12 out of 30 cases, demonstrating its efficacy in navigating the complexities of underground mine environments. Volumetric methods also contribute significantly, representing 11 out of 30 cases. The combined use of SLAM and volumetric methods, complemented by various neural network solutions detailed in the table, holds significant promise in influencing the future trajectory of 3D reconstruction methodologies.

Moreover, the research landscape extends its relevance to large-scale 3D reconstruction in underground mines, acknowledged by 23 out of 30 studies demonstrating promising applicability for large-scale 3D reconstruction, mainly showcasing the potential for effectively reconstructing expansive underground mines. This suggests a growing interest and feasibility in employing advanced techniques in large-scale subterranean environments.

However, the limited attention provided to robustness in dynamic scenes or objects (only 8 out of 30) and deformable objects (also 8 out of 30) highlights areas for improvement in current research efforts. Overall, the prevalence of RGB-D sensors, coupled with the widespread adoption of SLAM and volumetric methods, positions them as the optimal combination for addressing the challenges of 3D reconstruction in underground mining environments.

**Table 7.** A summary of studies conducted on real-time 3D reconstruction for underground mines.

| Author | Input Type | | | | Method | | | Scene | | Scalability | | Robustness to Dynamic Scenes/Object | | Robustness to Deformable Object | | Algorithms/Technique |
|---|---|---|---|---|---|---|---|---|---|---|---|---|---|---|---|---|
| | RGB | RGB-D | Other | SLAM | Volumetric | Sparse | Other | Indoor | Outdoor | Small | Large | Robust | Limited | Robust | Limited | |
| A. Brasoveanu et al. [93] | - | ✓ | - | - | ✓ | - | - | ✓ | - | - | ✓ | - | ✓ | - | ✓ | End-to-end, SIFT |
| B. Petit et al. [94] | ✓ | - | - | - | - | ✓ | - | ✓ | - | - | ✓ | - | ✓ | - | ✓ | EPVH |
| C. Zhao et al. [95] | - | ✓ | - | ✓ | - | - | - | ✓ | - | - | ✓ | - | ✓ | - | ✓ | CRF, CRF-RNN (recognition) |
| J. Lin et al. [96] | - | ✓ | - | ✓ | - | ✓ | - | ✓ | - | - | ✓ | - | ✓ | - | ✓ | Optimized Feature-adaptive |
| J. hua Lin et al. [97] | ✓ | - | - | ✓ | ✓ | - | Hybrid | ✓ | - | - | ✓ | - | ✓ | - | ✓ | CPU-to-GPU processing |
| A. Agudo et al. [98] | ✓ | - | - | - | - | ✓ | - | ✓ | ✓ | ✓ | - | - | ✓ | ✓ | - | EKF-FEM-FRP |
| Y. Xu et al. [99] | ✓ | - | - | ✓ | - | - | - | ✓ | ✓ | ✓ | - | - | ✓ | - | - | cGAN |
| M. Runz et al. [100] | - | ✓ | - | ✓ | - | - | - | ✓ | - | - | ✓ | ✓ | - | ✓ | - | Multi-model SLAM-based, Mask-RCNN |
| F. Lu et al. [101] | - | ✓ | - | ✓ | - | - | - | ✓ | - | - | ✓ | ✓ | - | ✓ | - | 6D pose prediction, GLSL |
| P. Stotko et al. [102] | - | ✓ | - | SLAMCast | - | - | - | ✓ | - | - | ✓ | - | ✓ | - | ✓ | Improved voxel block hashing |
| T. Laidlow et al. [103] | | ✓ | - | CNN-SLAM | - | - | - | ✓ | - | - | ✓ | - | ✓ | - | ✓ | Improved U-Net |
| C. Li et al. [104] | ✓ | - | RGB+IMU | - | ✓ | - | - | ✓ | ✓ | ✓ | - | - | ✓ | - | ✓ | MSCKF, EKF |
| M. Gong et al. [105] | ✓ | - | Lidar | - | ✓ | - | - | ✓ | ✓ | - | ✓ | - | ✓ | - | ✓ | SLAM |
| S. Zhang et al. [106] | - | ✓ | - | - | - | - | Hybrid | ✓ | - | - | ✓ | - | ✓ | - | ✓ | Pyramid FAST, Rotated BRIEF, GMD-RDS, improved RANSAC |
| Y. He et al. [107] | - | - | Lidar | - | ✓ | - | - | ✓ | - | ✓ | - | - | ✓ | - | ✓ | Improved voxel block hashing |
| Y. Fu et al. [108] | - | ✓ | - | ✓ | - | - | - | ✓ | - | - | ✓ | - | ✓ | - | ✓ | PEAC, AHC, ICP |
| C. Fei et al. [109] | - | - | SWIR | - | - | - | FTP | ✓ | - | ✓ | - | ✓ | - | ✓ | - | Improved FTP |
| D. Menini et al. [110] | - | ✓ | - | - | - | - | NN | ✓ | - | - | ✓ | - | ✓ | - | ✓ | AdapNet++, SGD optimizer |
| H. Matsuki et al. [111] | - | ✓ | - | ✓ | - | - | - | ✓ | - | - | ✓ | - | ✓ | - | ✓ | Sparse SLAM |
| Q. Jia et al. [112] | - | ✓ | - | - | - | - | YOLACT++ | ✓ | - | ✓ | - | - | ✓ | - | ✓ | YOLACT++, BCC-Drop, VJTR |
| S. Yu et al. [113] | ✓ | - | - | - | - | - | EC-PCS | - | ✓ | - | ✓ | - | ✓ | - | ✓ | Crowdsourcing (EC-PCS) |
| J. Sun et al. [114] | - | ✓ | - | ✓ | ✓ | - | Hybrid | ✓ | - | - | ✓ | - | ✓ | - | ✓ | FBV, GRU Fusion, ScanNet |
| S. Izadi et al. [115] | - | ✓ | - | - | ✓ | - | - | ✓ | - | - | ✓ | ✓ | - | ✓ | - | SLAM, TSDF, ICP |

**Table 7.** *Cont.*

| Author | Input Type | | | | Method | | | Scene | | Scalability | | Robustness to Dynamic Scenes/Object | | Robustness to Deformable Object | | Algorithms/Technique |
|---|---|---|---|---|---|---|---|---|---|---|---|---|---|---|---|---|
| | RGB | RGB-D | Other | SLAM | Volumetric | Sparse | Other | Indoor | Outdoor | Small | Large | Robust | Limited | Robust | Limited | |
| M. Keller et al. [116] | - | ✓ | - | - | ✓ | - | - | ✓ | - | - | ✓ | ✓ | - | ✓ | - | ICP |
| M. Niesner et al. [117] | - | ✓ | - | - | ✓ | - | - | ✓ | ✓ | - | ✓ | - | ✓ | - | ✓ | TSDF, ICP, Voxel Hashing, DDA |
| H. Kim et al. [118] | ✓ | - | - | ✓ | - | - | - | ✓ | ✓ | - | ✓ | ✓ | - | - | ✓ | EKF |
| A. Geiger et al. [119] | ✓ | - | - | - | - | ✓ | - | ✓ | ✓ | - | ✓ | ✓ | - | ✓ | - | SAD, NMS, KF, RANSAC, ELAS [120] |
| V. Pradeep et al. [121] | ✓ | - | - | - | ✓ | - | - | ✓ | - | ✓ | - | - | ✓ | - | ✓ | ZNCC, RANSAC, SVD, FAST, SAD, SDF |
| M. Zeng et al. [122] | - | ✓ | - | - | ✓ | - | - | ✓ | - | - | ✓ | ✓ | - | - | ✓ | TSDF, ICP |

## 4. Discussion

Mining operations are inherently hazardous, presenting various risks such as rock falls, equipment accidents, and challenging environments in both underground and surface mines. To address these risks, teleoperation and fully autonomous systems have emerged as promising solutions, enabling remote operation and monitoring of equipment. However, the broad adoption of these systems faces hindrances like implementation costs, technological complexity, safety concerns, operator training, reliability, industry resistance to change, and insufficient operator situational awareness. Current teleoperation systems rely on multiple monitors, leading to inefficiency and operator discomfort due to cognitive tunneling and depth detection issues. An obstacle to the widespread adoption of current teleoperation systems is the operator's reluctance due to the complexity of focusing on multiple video streams from various equipment cameras, making simple operations challenging. Leveraging 3D reconstruction techniques, these systems can move beyond the limitations of multi-monitor visualization, transitioning towards a virtual reality mode that significantly enhances an operator's awareness of their surroundings. The proposed platform aims to address this issue by combining multiple views into a VR goggle.

Choosing appropriate sensors and algorithms for 3D reconstruction used in VR teleoperation platforms is an intricate task that requires careful consideration of the trade-offs between precision, speed, and flexibility. This discussion explores the challenges of the selection process and outlines potential solutions proposed in recent research. The rapid real-time data processing requirement complicates the intricate task of selecting sensors and algorithms for 3D reconstruction. Achieving speed without sacrificing accuracy poses a challenge, as not all sensors and algorithms can meet this demand. The unpredictable surface and underground mine environments also introduce an extra layer of complexity to the selection process.

Current studies thoroughly explore the effectiveness of different sensor types and algorithms in 3D reconstruction. Some studies highlight the prevalent utilization of RGB and RGB-D sensors in real-time indoor 3D visualization methods. These sensors are particularly adept at simultaneously capturing color and depth information, making them highly suitable for dynamic underground mine environments. These studies also shed light on the limited use of specific sensor types like binoculars, spherical cameras, radar, and lidar within underground mine environments. These findings encourage further investigation into the potential benefits that these sensors might provide, especially in non-real-time applications where considerations like lighting conditions and texture significantly influence outcomes. In surface mine environments, some studies recognize the widespread use of RGB cameras in conjunction with a fusion of lidar as the predominant sensor combination. This deliberate fusion strikes a harmonious balance between the comprehensive visual data captured by cameras and the depth information supplied by lidar, making it well suited for real-time and non-real-time 3D visualization in open pit mines.

Some studies emphasize the significance of depth-based algorithms in real-time 3D reconstruction within underground mines. Coupled with sensors like RGB-D cameras, these algorithms have emerged as the most employed methods. The depth information they furnish substantially improves the precision of reconstructions, making them particularly suitable for applications where accuracy is of utmost importance. Point cloud-based algorithms for 3D reconstruction have become the leading methods for non-real-time surface mining applications. Utilizing data from lidar sensors or photogrammetry, these algorithms meticulously craft a detailed representation of a scene by processing extensive sets of individual data points in three-dimensional space. Their widespread adoption of non-real-time scenarios is justified by their proficiency in capturing intricate details in diverse landscapes or large structures. Point cloud-based approaches excel in managing irregular terrains, delivering a comprehensive and accurate portrayal of surface mine scenes. Their versatility in handling various data sources and navigating the intricacies of open pit environments establishes them as the preferred choice for researchers and

practitioners involved in non-real-time outdoor 3D reconstruction projects, prioritizing precision and comprehensive detail.

A key obstacle in 3D reconstruction involves the significant computational power needed for the timely processing and reconstruction of three-dimensional scenes. Analyzing extensive data from various sensors comes with complexities that require considerable computing resources, frequently surpassing the capabilities of typical hardware setups. In response to this challenge, continuous research is directed towards creating and refining algorithms for 3D reconstruction that achieve a harmonious balance between computational complexity and precision. Furthermore, progress in parallel processing, the adoption of specialized hardware such as Graphics Processing Units (GPUs), and the exploration of cloud-based computing solutions all play a role in improving the computational efficiency of 3D reconstruction procedures. These initiatives strive to fully exploit the capabilities of contemporary computing technologies fully, thereby making real-time or near-real-time 3D reconstruction more attainable and practical across various applications, spanning from VR teleoperation to autonomous systems.

One notable criticism of VR platforms is the occurrence of "motion sickness" or simulator sickness [123,124]. This problem can be attributed to factors such as visual–vestibular mismatch, frame rate, latency, field of view, rapid locomotion, and inner ear sensitivity [125]. These issues can be significantly mitigated by optimizing hardware settings, increasing frame rates, reducing latency, limiting field of view, and employing high-detail models [126]. Enhancements in 3D reconstruction for mining teleoperation can also positively impact various other research areas in mining, including autonomous systems, slope stability monitoring, resource estimation, the digital twin concept, geological analysis, and AI integration.

Incorporating VR teleoperation systems is a recently introduced idea in the mining sector. Despite the clear benefits regarding safety and operational efficiency, there has been some hesitancy in the industry, primarily due to apprehensions about the related expenses. Nevertheless, it is important to acknowledge that the initial investment in these systems constitutes a forward-thinking approach with substantial potential for enhancing safety and overall progress in the mining industry. This reduces immediate safety hazards for operators and offers enduring advantages by decreasing the frequency of accidents and injuries. As technology advances and costs potentially decline with broader acceptance, the mining industry is poised to reap significant benefits in terms of enhanced worker safety, streamlined operational processes, and heightened overall productivity.

## 5. Conclusions

The 3D reconstruction of environments has garnered significant interest due to its diverse range of applications, yet its adoption within the mining industry remains relatively limited, not fully achieving its potential. This research aimed to explore how 3D reconstruction can extend beyond conventional applications in mine surveying, potentially revolutionizing fully automated and teleoperation systems for mining equipment. The study delved into major scientific studies and manufacturing companies that exhibit potential for conversion to VR mode within teleoperation technology. Also, it has focused on identifying and evaluating various sensors used for 3D reconstruction in underground and surface mining operations, categorizing them into passive and active sensors. Additionally, this study investigated the scope of existing 3D reconstruction research studies applicable to surface and underground environments, emphasizing research with potential utility in the mining industry. Moreover, it differentiated the non-real-time process of surface 3D reconstruction, primarily utilizing point clouds based on structure-from-motion algorithms. Furthermore, it dissected real-time 3D reconstruction within underground environments, elucidating these systems' specific frameworks, algorithms, sensors, and practical applications. Finally, the paper scrutinizes the challenges and advantages of current teleoperation platforms in mining and examines how VR mode integration could address these issues.

**Author Contributions:** Conceptualization, A.K.-P. and J.S.; methodology, A.K.-P. and A.M.-M.; software, A.K.-P.; visualization A.K.-P.; investigation, A.K.-P.; writing—original draft, A.K.-P.; writing—review and

editing, A.M.-M.; supervision, J.S. All authors have read and agreed to the published version of the manuscript.

**Funding:** Center for Disease Control and Prevention and the National Institute for Occupational Health and Safety under contract number 75D30119C06044.

**Institutional Review Board Statement:** Not applicable.

**Informed Consent Statement:** Not applicable.

**Data Availability Statement:** Not applicable.

**Acknowledgments:** This study is a part of the project supported by the Center for Disease Control and Prevention and the National Institute for Occupational Health and Safety. The authors are responsible for this project's recommendations, conclusions, and findings. The outcomes do not necessarily reflect the views of the National Science Foundation, the Center for Disease Control and Prevention, or the National Institute for Occupational Health and Safety.

**Conflicts of Interest:** The authors declare no conflicts of interest.

## Abbreviations

| | |
|---|---|
| ANMS | Adaptive non-maximal suppression |
| AR | Augmented reality |
| BRIEF | Binary robust independent elementary features |
| CGAN | Conditional generative adversarial nets |
| CMVS | Clustering views for multiview stereo |
| CRF | Conditional random fields |
| DBSCAN | Density-based spatial clustering of applications with noise |
| DDA | Amanatides and Woo 1987 |
| EKF | Extended Kalman filter |
| FEM | Finite element method |
| FRP | Free rigid priors |
| GLSL | OpenGL shading language |
| GMD-RDS | Gaussian mixture distribution—random down-sampling |
| ICP | Iterative closest point |
| IMU | Inertial measurement unit |
| MSCKF | Multi-state constraint Kalman filter |
| PEAC | Plane extraction in organized point clouds using agglomerative hierarchical clustering |
| VIO | Visual–inertial odometry |
| VR | Virtual reality |

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
