# Peer review of "Applications of 3D Reconstruction in Virtual Reality-Based Teleoperation: A Review in the Mining Industry"

_technologies, doi:10.3390/technologies12030040_

Round 1
Reviewer 1 Report
Comments and Suggestions for Authors
Although multi-view platforms have enhanced work efficiency in mining teleoperation systems, they also induce "cognitive tunneling" and depth-detection issues for operators. This study investigates the existing gap in fully immersive teleoperation systems within the mining industry, aiming to identify the most optimal methods for their development and ensuring operator's safety. It is a carefully done study and the findings are of considerable interest. However, I thought it still has some deficiencies and I recommend to a major revision before acceptable publication. Detailed comments are listed below:
1. A more detailed description of previous research in introduction is needed, existed works on mobile robot navigation (Velocity-free localization of autonomous driverless vehicles in underground intelligent mines. IEEE Transactions on Vehicular Technology, 2020, 69(9): 9292-9303. Control of velocity-constrained stepper motor-driven hilare robot for waypoint navigation. Engineering, 2018, 4(4): 491-499.) could be helpful to give some useful information about the intelligent development of mining industry. Authors are encouraged to summarize these works for providing comprehensive content to readers of your paper.
2. How to achieve real scene simulation and real-time interaction using virtual reality technology in mining practice, and whether operators are in line with specific technical requirements. please make it clearly.
3. Is the data on the gradual decrease in the number of deaths from mining operations credible? With the rapid development of industrial technology, the mining industry is of utmost importance, and only in recent years have intelligent technologies been applied to the mining industry. Also, do death data refer to metal mines or non-metal mines?
4. Is there a corresponding relationship between Figure 1 and Figure 2? Please elaborate in detail on the relationship between the popularization of intelligent technology and personnel death, and is it directly proportional? Is there any relevant data to verify? Please make it clearly.
5. Menzi Muck M545 can learn and work independently without worker intervention, How is the initial state set? How to ensure the reliability of unmanned operation?
6. Intelligent excavators have a series of advantages such as high efficiency, strong flexibility, and easy operation, but their disadvantages include high price, high maintenance costs, and high operator requirements. How should users consider comprehensively when making choices?
7. Is the development of equipment just a stacking of multiple hardware? How to use algorithms to carry out unmanned operation of intelligent devices during intelligent control?
8. The application of sensors is crucial. What types of sensors are included in the equipment development process? What kind of role did each of them play? Has the author considered a tabular presentation?
Comments on the Quality of English LanguageMinor editing of English language required
Author Response
Author's Response to Review Comments
We would like to express our gratitude to the reviewers for their insightful comments and meticulous review of our manuscript. We have diligently addressed each comment, making the necessary modifications and technical corrections. As a result, we believe the paper's readability and quality have been significantly enhanced. We have also provided detailed point-by-point responses to the comments.
Reviewer 1:
- A more detailed description of previous research in introduction is needed, existed works on mobile robot navigation (Velocity-free localization of autonomous driverless vehicles in underground intelligent mines. IEEE Transactions on Vehicular Technology, 2020, 69(9): 9292-9303. Control of velocity-constrained stepper motor-driven hilare robot for waypoint navigation. Engineering, 2018, 4(4): 491-499.) could be helpful to give some useful information about the intelligent development of mining industry. Authors are encouraged to summarize these works for providing comprehensive content to readers of your paper.
Response:
The papers were studied and incorporated into the introduction section. The first paragraph of the introduction has been updated, and we applied these references to the text to improve the paper's quality.
- How to achieve real scene simulation and real-time interaction using virtual reality technology in mining practice, and whether operators are in line with specific technical requirements. please make it clearly.
Response:
- Response to the first question: The real-time scene simulation is done using techniques explained in the “3.2.2. 3D reconstruction in surface mines” section for surface mines and “3.2.3. 3D reconstruction in underground mines” section for underground mines.
- Response to the second question: VR teleoperation is performed in the same manner as regular teleoperation. However, the conventional visualization type is replaced with a 3D reconstruction of the environment.
- Is the data on the gradual decrease in the number of deaths from mining operations credible? With the rapid development of industrial technology, the mining industry is of utmost importance, and only in recent years have intelligent technologies been applied to the mining industry. Also, do death data refer to metal mines or non-metal mines?
Response:
- Response to the first question: Yes. The data were collected from the websites of the "Mine Safety and Health Administration (MSHA)" and the "National Institute for Occupational Safety and Health (NIOSH)."
The U.S. Department of Labor's Mine Safety and Health Administration (MSHA) helps to reduce deaths, injuries, and illnesses in the nation's mines with a variety of activities and programs. The Agency develops and enforces safety and health rules for all U.S. mines, and provides technical, educational, and other types of assistance to mine operators. MSHA works cooperatively with industry, labor, and other federal and state agencies to improve safety and health conditions for all miners in the United States “Website Link: https://www.msha.gov”
The National Institute for Occupational Safety and Health (NIOSH) conducts research and makes recommendations for the prevention of work-related injury and illness “Website Link: https://www.cdc.gov/niosh/”
- Response to the second question: The data includes both metal mines and non-metal mines.
- Is there a corresponding relationship between Figure 1 and Figure 2? Please elaborate in detail on the relationship between the popularization of intelligent technology and personnel death, and is it directly proportional? Is there any relevant data to verify? Please make it clearly.
Response:
- Response to the first question: Yes. Figure 1 indicates the number of injuries, including fatalities, in mining operations from 2000 to 2022, while Figure 2 indicates the number of powered haulage fatalities in mining operations during the same period. Figure 2 data is a subset of Figure 1, meaning Figure 1 includes Figure 2.
- Response to the second question: We have not found any specific study or data illustrating the exact relationship between the popularization of intelligent technology and personnel deaths in the mining industry.
- Menzi Muck M545 can learn and work independently without worker intervention, How is the initial state set? How to ensure the reliability of unmanned operation?
Response:
The 'HEAP' project, conducted in a Zurich laboratory, involved customizing the 'Menzi Muck M545' excavator for two purposes: automation and teleoperation. Our study employed the project as an illustrative example of a teleoperated excavator. I acknowledge that the unmanned operation aspect is not in the scope and objectives of our manuscript.
- Intelligent excavators have a series of advantages such as high efficiency, strong flexibility, and easy operation, but their disadvantages include high price, high maintenance costs, and high operator requirements. How should users consider comprehensively when making choices?
Response:
The primary focus of the paper is on "Applications of 3D Reconstruction in Virtual Reality-Based Teleoperation." The discussion emphasizes techniques for 3D reconstruction applicable to teleoperation, and the research scope extends beyond equipment types, such as excavators or intelligent excavators.
The contributions of the research in the introduction section have been modified to address any confusion.
- Is the development of equipment just a stacking of multiple hardware? How to use algorithms to carry out unmanned operation of intelligent devices during intelligent control?
Response:
Importantly, this research does not delve into equipment development. Instead, it concentrates on discussing techniques and algorithms for 3D reconstruction of the environment.
- The application of sensors is crucial. What types of sensors are included in the equipment development process? What kind of role did each of them play? Has the author considered a tabular presentation?
Response:
The sensors highlighted in section "3.2.1. Data Acquisition for 3D Reconstruction" are specifically utilized for data acquisition in the 3D reconstruction of environments, not for equipment development.
Reviewer 2 Report
Comments and Suggestions for Authors
In this manuscript, the authors simply state that the review explores 3D reconstruction applications in virtual reality teleoperation for mining. While multi-view platforms enhance efficiency, they can cause cognitive tunneling. Fully immersive VR addresses this, but its implementation poses challenges. The study assesses methods for developing immersive systems, focusing on visualization types and 3D reconstruction techniques, particularly in underground mining. As emphasized by the authors, teleoperation systems can be employed as one of the most promising solutions to address these challenges, but they are not widely used because of a lack of efficiency.
This manuscript is extremely comprehensive with adequate references and citations. A variety of techniques are discussed and analyzed in detail. Ample figures, tables, and graphs are presented to assist readers/researchers to get engaged and even inspired. Overall, this manuscript is well-designed and organized. Minor grammar check and spelling are helpful.
Great work!
Comments on the Quality of English LanguageMinor
Author Response
Author's Response to Review Comments
We would like to express our gratitude to the reviewers for their insightful comments and meticulous review of our manuscript. We have diligently addressed each comment, making the necessary modifications and technical corrections. As a result, we believe the paper's readability and quality have been significantly enhanced. We have also provided detailed point-by-point responses to the comments.
Reviewer 2:
This manuscript is extremely comprehensive with adequate references and citations. A variety of techniques are discussed and analyzed in detail. Ample figures, tables, and graphs are presented to assist readers/researchers to get engaged and even inspired. Overall, this manuscript is well-designed and organized. Minor grammar check and spelling are helpful.
Response:
We have reviewed the whole manuscript and corrected all the spelling and grammar mistakes.
Round 2
Reviewer 1 Report
Comments and Suggestions for Authors
This is the second time this manuscript is being reviewed. In the first review, several suggestions were made to enable the Authors enhance the quality of the paper. The Authors have diligently responded to the suggestions and made changes to the content of the paper. The quality of the paper was improved.I am satisfied that the paper is acceptable for publication.
Comments on the Quality of English LanguageMinor editing of English language required